# The nucleosomal barrier to promoter escape by RNA polymerase II is overcome by the chromatin remodeler Chd1

Peter J Skene[1], Aaron E Hernandez[1,2], Mark Groudine[1,3], Steven Henikoff[1,2]*

[1]Basic Sciences Division, Fred Hutchinson Cancer Research Center, Seattle, United States; [2]Howard Hughes Medical Institute, Fred Hutchinson Cancer Research Center, Seattle, United States; [3]Department of Radiation Oncology, University of Washington School of Medicine, Seattle, United States

**Abstract** RNA polymerase II (PolII) transcribes RNA within a chromatin context, with nucleosomes acting as barriers to transcription. Despite these barriers, transcription through chromatin in vivo is highly efficient, suggesting the existence of factors that overcome this obstacle. To increase the resolution obtained by standard chromatin immunoprecipitation, we developed a novel strategy using micrococcal nuclease digestion of cross-linked chromatin. We find that the chromatin remodeler Chd1 is recruited to promoter proximal nucleosomes of genes undergoing active transcription, where Chd1 is responsible for the vast majority of PolII-directed nucleosome turnover. The expression of a dominant negative form of Chd1 results in increased stalling of PolII past the entry site of the promoter proximal nucleosomes. We find that Chd1 evicts nucleosomes downstream of the promoter in order to overcome the nucleosomal barrier and enable PolII promoter escape, thus providing mechanistic insight into the role of Chd1 in transcription and pluripotency.

*For correspondence: steveh@fhcrc.org

**Competing interests:** The authors declare that no competing interests exist.

**Reviewing editor**: Joaquin M Espinosa, HHMI/University of Colorado at Boulder, United States

## Introduction

The eukaryotic genome is packaged into chromatin, with the basic unit being a nucleosome, consisting of 147 base pairs (bp) of DNA wrapped around an octamer of histone proteins. Chromatin limits the accessibility of DNA-binding factors to the underlying DNA sequence and presents a strong physical barrier that must be overcome by RNA polymerase II (PolII) during transcription. In vitro experiments have shown that transcription initiation is the most affected and that elongation is also highly inefficient unless nucleosomes are destabilized (*Lorch et al., 1987*; *Hodges et al., 2009*; *Jin et al., 2010*; *Bintu et al., 2012*). Despite this barrier, in vivo elongation rates through chromatin are comparable to rates on naked DNA, indicating that the cell has evolved mechanisms to overcome this chromatin barrier (*Li et al., 2007*; *Ardehali and Lis, 2009*; *Singh and Padgett, 2009*; *Petesch and Lis, 2012*).

Through the combined action of PolII, elongation factors, nucleosome modifications and chromatin remodelers, the chromatin barrier to transcription is highly dynamic, with nucleosomal turnover correlating with gene expression (*Deal et al., 2010*). Nucleosomes are disrupted to allow efficient elongation and are reassembled in the wake of PolII to prevent cryptic initiation from intragenic sequences (*Cheung et al., 2008*; *Petesch and Lis, 2008*; *Quan and Hartzog, 2010*). In metazoans, PolII is often enriched immediately downstream of the transcription start site (TSS), raising the possibility that this is a significant barrier to promoter escape in the progression from transcriptional initiation to elongation, with an unknown mechanism by which this barrier is overcome (*Guenther et al., 2007*; *Mavrich et al., 2008*; *Schones et al., 2008*).

**eLife digest** DNA is tightly packaged in a material called chromatin inside the cell nucleus. To produce proteins this DNA must first be transcribed to produce a molecule of messenger RNA, which is then translated to make a protein. To assist with this process cells 'unpack' certain regions of the DNA so that enzymes that catalyze the different steps in this process can have access to the DNA.

A protein called Chd1 is involved in the unpacking process in yeast, but its role in more complex animals is not clear. Now, Skene et al. have shown that this protein is needed to allow the enzyme that catalyzes the transcription of DNA—an enzyme called RNA polymerase II—to do its job. Chd1 acts to unpack the tightly packaged DNA from chromatin, thus allowing the transcription of the DNA to proceed. In the absence of Chd1 activity, RNA polymerase II stalls at the gene promoter—the region of DNA that starts the transcription of a particular gene. This work highlights how the packaging of DNA in the cell is highly dynamic and controls fundamental biological processes.

Skene et al. modified a well-known genetic technique called ChIP-seq. Previous ChIP-seq protocols typically provided a blurry, low-resolution map of where proteins bound to chromatin. Skene et al. used an enzyme to 'chew back' the DNA to reveal the exact 'footprints' of the Chd1 protein and the RNA polymerase II enzyme on the chromatin in mice. It will be possible to adapt this new protocol to map the positions of other proteins, which will help to improve our understanding of the ways in which chromatin regulates access to DNA.

Although the intrinsic structural preferences of DNA contribute substantially to nucleosome occupancy and positioning in vivo, this chromatin landscape is further manipulated by chromatin remodelers (*Segal and Widom, 2009*; *Kaplan et al., 2010*). Chromatin remodelers use the energy from ATP hydrolysis to evict, assemble, and slide nucleosomes in order to guide the proper nucleosome positioning at key sites, such as eukaryotic promoters, where they maintain the nucleosome-depleted region (NDR) and phased flanking nucleosomes (*Hartley and Madhani, 2009*; *Gkikopoulos et al., 2011*; *Tolkunov et al., 2011*). Chromatin remodelers share a common Snf2 helicase-like ATPase domain that is required for the disruption of histone–DNA contacts (*Hota and Bartholomew, 2011*; *Petty and Pillus, 2013*). Furthermore, they can be separated into two groups based upon their requirement for extra-nucleosomal DNA. First, ISWI, INO80/SWR1 and CHD require extra-nucleosomal DNA to remodel nucleosomes, whereby they are primarily involved in nucleosome assembly and generating equally spaced arrays by binding to stretches of exposed DNA and shifting nucleosomes into the gap (*Whitehouse et al., 2003*; *McKnight et al., 2011*; *Udugama et al., 2011*). Second, SWI/SNF and RSC remodelers slide or evict nucleosomes irrespective of extra-nucleosomal DNA (*Whitehouse et al., 1999*; *Lorch et al., 2011*).The potential role of remodelers in overcoming the nucleosomal barrier has not been fully investigated.

Studies in yeast have suggested the main role for the chromatin remodeler Chd1 to be within the gene body, where Chd1 reassembles and positions nucleosomes in order to prevent cryptic initiation (*Cheung et al., 2008*; *Gkikopoulos et al., 2011*; *Radman-Livaja et al., 2012*; *Smolle et al., 2012*; *Zentner et al., 2013*). However, the promoter chromatin architecture is different in metazoans, with the TSS embedded in the nucleosome-depleted region, raising the possibility that this differing chromatin landscape contributes to metazoan PolII accumulating to high levels ahead of the first nucleosome (*Mavrich et al., 2008*; *Schones et al., 2008*; *Rahl et al., 2010*). Here, we show that mammalian Chd1 controls nucleosome dynamics at actively transcribed genes. We find that Chd1 is responsible for the vast majority of PolII-directed nucleosome turnover at the promoter and is required to allow efficient PolII promoter escape, with PolII becoming stalled in the absence of Chd1 activity. Stalling with the loss of Chd1 activity implies that Chd1 is required to overcome the nucleosomal barrier to allow transcription within a chromatin context. In contrast, in the gene body, Chd1 reassembles nucleosomes and suppresses histone turnover. Interestingly, mammalian Chd1 is required for pluripotency of embryonic stem (ES) cells by maintaining euchromatin and is also required for efficient reprogramming, suggesting that Chd1 plays a key role in mammalian cellular identity (*Gaspar-Maia et al., 2009*). Through both positively and negatively impacting histone dynamics, Chd1 plays roles in transcription and in regulating pluripotency and reprogramming.

# Results

## Chd1 is recruited to the promoter and its ATPase activity is required for binding to extend into the gene body

To study the role of mammalian Chd1, mouse embryonic fibroblasts (MEFs) were transfected with a transgene expressing FLAG-tagged wild-type mouse Chd1 or a mutant variant harboring the replacement of a conserved lysine to an arginine residue (K510R) in the ATP-binding site of the remodeler (*Figure 1A*). The K510R Chd1 mutation eliminates the catalytic activity without disrupting its ability to interact with other proteins, thereby generating a dominant negative (*Corona et al., 2004*). Mutation of this conserved lysine residue has been used to functionally investigate various chromatin remodelers including Brahma and ISWI in *Drosophila* (*Figure 1A*; *Elfring et al., 1998*; *Deuring et al., 2000*). In comparison to knocking down Chd1 expression, a dominant negative approach is less likely to enable redundant mechanisms to mask protein functions. Western analysis indicated that the FLAG-tagged proteins migrated at the expected size and were only moderately over-expressed relative to endogenous Chd1 in the MEFs (*Figure 1B*).

First, we determined the genomic localization of Chd1 using chromatin immunoprecipitation in combination with high throughput sequencing (ChIP-seq). We found that under native conditions only 14% of Chd1 could be extracted (*Figure 1—figure supplement 1A*). This led us to pursue a formaldehyde cross-linked ChIP strategy, which allows for harsher extraction conditions. Sonication is often used in ChIP to fragment the formaldehyde-cross-linked chromatin. This, however, only provides low resolution, as the fragment size is typically 100–300 bp in comparison to the footprint of either PolII or Chd1, which is 35 bp and 20–45 bp respectively (*Samkurashvili and Luse, 1996*; *Ryan et al., 2011*). We therefore used *Micrococcal* nuclease (MNase) digestion of cross-linked chromatin to increase the resolution of ChIP-seq. MNase digests unprotected DNA, leaving DNA fragments corresponding to mono-nucleosomes and sub-nucleosomal fragments, presumably protected by proteins cross-linked to the extra-nucleosomal DNA (*Figure 1—figure supplement 1B*). By using a modified Solexa library preparation protocol in combination with paired-end sequencing, we were able to map DNA fragments as small as ~35 bp (*Henikoff et al., 2011*). By using MNase to fragment the chromatin and minimal sonication we were able to achieve essentially complete extraction of chromatin and associated proteins, thereby overcoming a significant limitation of native ChIP (*Figure 1—figure supplement 1C*). This technique is highly transferable, as it requires no special tool development and uses current sequencing technology.

Initially analysis of all the recovered fragments was performed. As expected, wild-type Chd1 was recruited to the 5′ end of genes, similar to what has been shown for endogenous Chd1 in mouse ES cells (*Figure 1C,D*; *Gaspar-Maia et al., 2009*). Furthermore, the dominant negative K510R-Chd1 showed a broadly similar chromatin recruitment pattern, consistent with the mutation not affecting interactions with recruiting factors. In addition, we found that the enrichment of Chd1 was primarily just downstream of the TSS, with recruitment correlating with PolII occupancy of the promoter, as measured in the MEFs expressing wild-type Chd1 (*Figure 1D*). PolII has previously been shown to crosslink to nucleosomes (*Koerber et al., 2009*). Therefore, mapping the DNA recovered by immunoprecipitation of these cross-linked PolII:nucleosome complexes will not identify the exact binding site of PolII on DNA. To precisely map the position of PolII and Chd1, we mapped 35–75 bp fragments, which more closely corresponds to the footprint of PolII and Chd1 (*Figure 1—figure supplement 2A*). For PolII, there was a shift of the maximal occupancy towards the TSS, from +150 bp for all fragment sizes to a peak centered over the TSS, likely indicating that the larger fragments correspond to PolII crosslinked to the +1 nucleosome. Similarly, for both wild-type and K510R Chd1, larger size classes indicated Chd1 crosslinked to the well-positioned +1 nucleosome. Analysis of short fragments, however, indicated wild-type Chd1 was enriched over the beginning of the gene body with a steady decline from +500 bp. However, in the absence of ATPase activity, K510R-Chd1 density peaked downstream of the TSS at +60 bp, where it accumulated to higher levels than wild-type Chd1 and was depleted in the gene body (*Figure 1E*, *Figure 1—figure supplement 2B*). To confirm that mapping of short fragments was similar to that of all fragments, genes were ranked by PolII promoter occupancy as determined from all fragment lengths and separated into quintiles (*Figure 1—figure supplement 2C*). We find a correlation between Chd1 recruitment and PolII occupancy, as seen for all fragment lengths. Overall, this novel ChIP-seq strategy based on the MNase digestion of cross-linked chromatin and the mapping of short fragments allows precise mapping of the footprint of PolII and Chd1. We find that Chd1 binds just

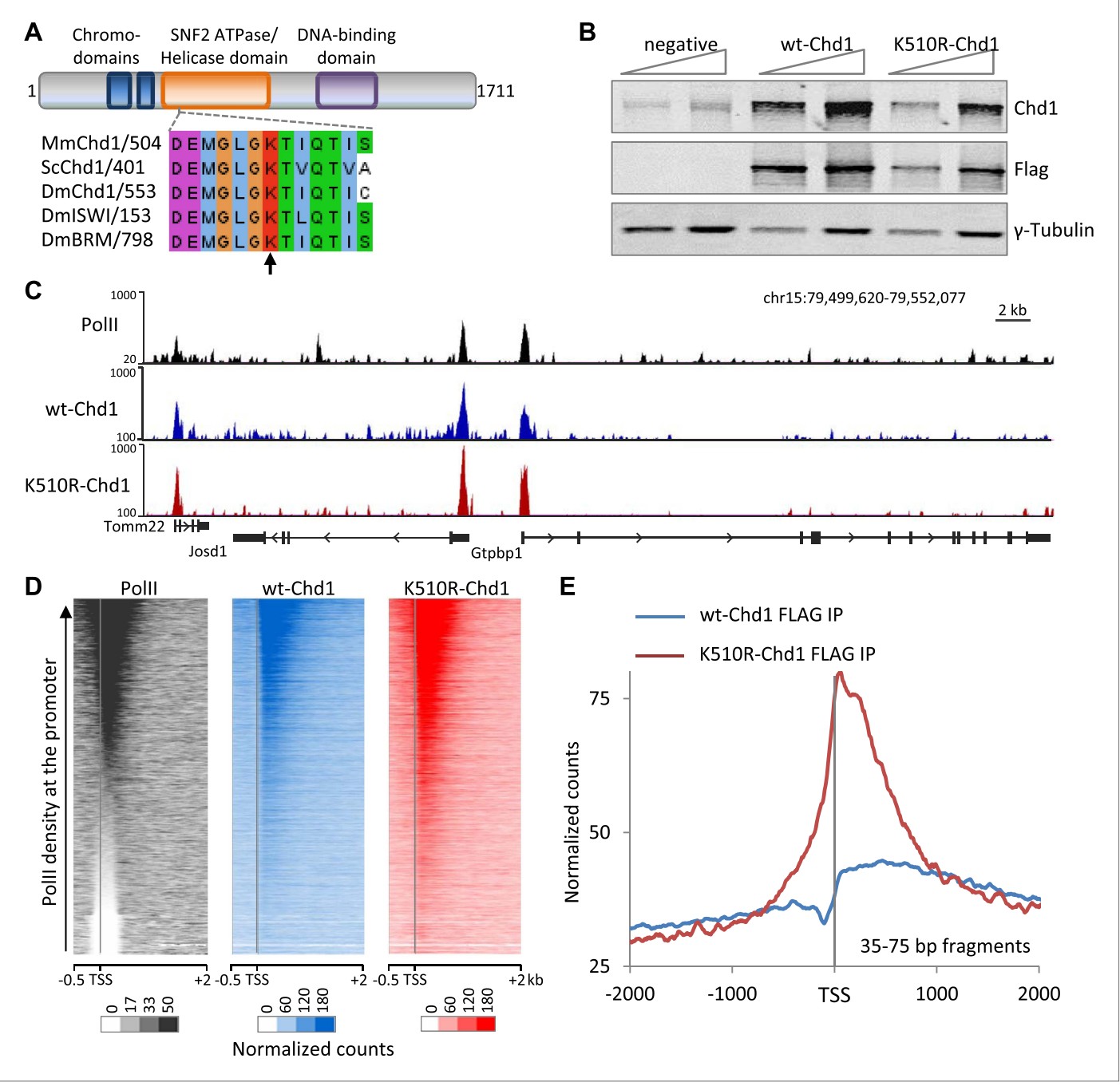

**Figure 1**. Chd1 is recruited to promoters with high PolII occupancy and requires ATPase activity to extend into the gene body. (**A**) Schematic of the domain structure of the full-length mouse Chd1 (1-1711) used to generate the N-terminal FLAG-tagged construct. A region corresponding to the Chd1 ATP-binding pocket is shown below and aligned to various homologues from *Saccharomyces cerevisiae* and *Drosophila melanogaster*. The arrow indicates the conserved lysine residue mutated to form the dominant negative. (**B**) MEFs stably expressing either FLAG-tagged wild-type Chd1 or the dominant negative K510R were subjected to western analysis with a two-fold dilution series. Untransfected MEFs were used as a reference. (**C**) A representative genome browser snapshot of ChIP-seq data (all fragment lengths) indicating the high occupancy of PolII, wildtype-Chd1 and K510R-Chd1 at gene promoters. PolII distribution was determined in cells expressing wildtype-Chd1 using the N20 antibody. Normalized counts are indicated on the y-axis. (**D**) Chd1 binding to the 5' end of genes was determined by ChIP-seq. Here, all recovered DNA fragments, irrespective of length, were analyzed. Each row of the heatmap represents the binding pattern across the −0.5 kb to +2 kb region flanking the TSS. Genes were ranked by the level of PolII occupancy in the −100 to +300 bp region, as measured by ChIP-seq in the MEFs expressing wildtype Chd1, using the N20 antibody that binds to the N-terminus of the largest subunit of PolII. (**E**) Analyzing only the short DNA fragments recovered (35–75 bp) allows precise

*Figure 1. Continued on next page*

*Figure 1. Continued*

mapping of Chd1, indicating that wildtype Chd1 binding tracks into the gene body, whereas K510R-Chd1 accumulates just downstream of the TSS. The genome-wide average is shown across the 4 kb encompassing the TSS.

The following figure supplements are available for figure 1:

**Figure supplement 1**. A novel crosslinking ChIP strategy allows near-complete extraction of chromatin-associated proteins and high-resolution mapping of binding sites.

**Figure supplement 2**. Analysis of short fragments provides high-resolution mapping of Chd1 and PolII binding sites at the promoter.

downstream of the TSS, correlates with PolII occupancy at the promoter and requires ATPase activity for the binding to extend into the gene body.

## Chd1 activity is required to maintain nucleosome occupancy within the promoter region and the gene body

Chromatin remodelers are known to have a major role in the correct positioning and occupancy of nucleosomes (*Gkikopoulos et al., 2011*). Therefore we determined the nucleosome profile in cells expressing either wild-type or K510R Chd1 by mapping mono-nucleosome-sized fragments recovered from the sequencing of MNase digested, cross-linked chromatin. Cells expressing wild-type Chd1 display the classical metazoan nucleosome organization pattern, with a pronounced nucleosome depleted region (NDR) containing the TSS, flanked by well positioned nucleosomes, with the +1 nucleosome entry site ~50 bp downstream of the TSS (*Figure 2A*; *Mavrich et al., 2008*; *Schones et al., 2008*). In addition, the position of the +1 nucleosome corresponded to the level of PolII promoter occupancy, with the 5′ edge progressively moving towards the TSS, to approximately +15 bp for genes with the lowest PolII occupancy, as previously indicated by ranking human nucleosome positions by steady-state RNA levels (*Schones et al., 2008*). The positioning of the nucleosomes was not affected by the loss of Chd1 activity, but the nucleosome occupancy was reduced both upstream and downstream of the TSS. In contrast to *Saccharomyces cerevisiae*, where deletion of Chd1 had no effect on the +1 nucleosome occupancy but only the surrounding nucleosomes, blocking Chd1 activity in MEFs affected promoter proximal nucleosomes including the +1 nucleosome (*Gkikopoulos et al., 2011*). In *S. cerevisiae*, the TSS is found just within the +1 nucleosome, thereby suggesting that pre-initiation complex formation requires loss of the +1 nucleosome (*Rhee and Pugh, 2012*). However, metazoans have a different promoter nuclear architecture with the TSS embedded in the nucleosome depleted region and the +1 nucleosome entry site ~50 bp downstream (*Mavrich et al., 2008*; *Schones et al., 2008*). Therefore, loss of the mammalian +1 nucleosome occupancy observed here likely reflects a unique role of mammalian Chd1 because of differences to *S. cerevisiae* in promoter architecture and perhaps the mechanism of transcription through the promoter proximal chromatin. Decreased nucleosome occupancy was also apparent throughout the gene body including upstream of the transcriptional end site (TES).

The decrease in nucleosome occupancy was most apparent at genes with the highest PolII levels at the promoter, as expected given that Chd1 occupancy correlates with PolII promoter occupancy, with no statistically significant change at genes with low PolII levels at the promoter (*Figure 2A*). Genes were ranked by wildtype-Chd1 binding within the beginning of the gene to more directly test the role of Chd1 in maintaining nucleosome occupancy (*Figure 2B*). Genes with the highest recruitment of Chd1 exhibited the most dramatic loss of nucleosomes at the promoter upon expression of K510R-Chd1, consistent with a direct role of Chd1. Additionally, genes with the lowest recruitment of wild-type Chd1 displayed a markedly different nucleosome landscape, with unphased nucleosomes and the absence of a pronounced NDR (*Figure 2B*). Despite this relatively flat profile at the TSS being insensitive to the expression of K510R-Chd1, it suggests that the 'open' and 'closed' promoter architecture, as defined by the existence of an NDR at the TSS, can be categorized based on recruitment of remodelers such as Chd1 (*Cairns, 2009*). The loss of nucleosome occupancy within the last 1 kb of the gene body was independent of the level of Chd1 recruitment to the promoter (*Figure 2B*). In contrast, the loss of nucleosome occupancy in the gene body correlated with level of Chd1 binding in the gene body (*Figure 2—figure supplement 1A*). This indicates that Chd1 action within the distal part of the gene body is independent of its recruitment to the promoter.

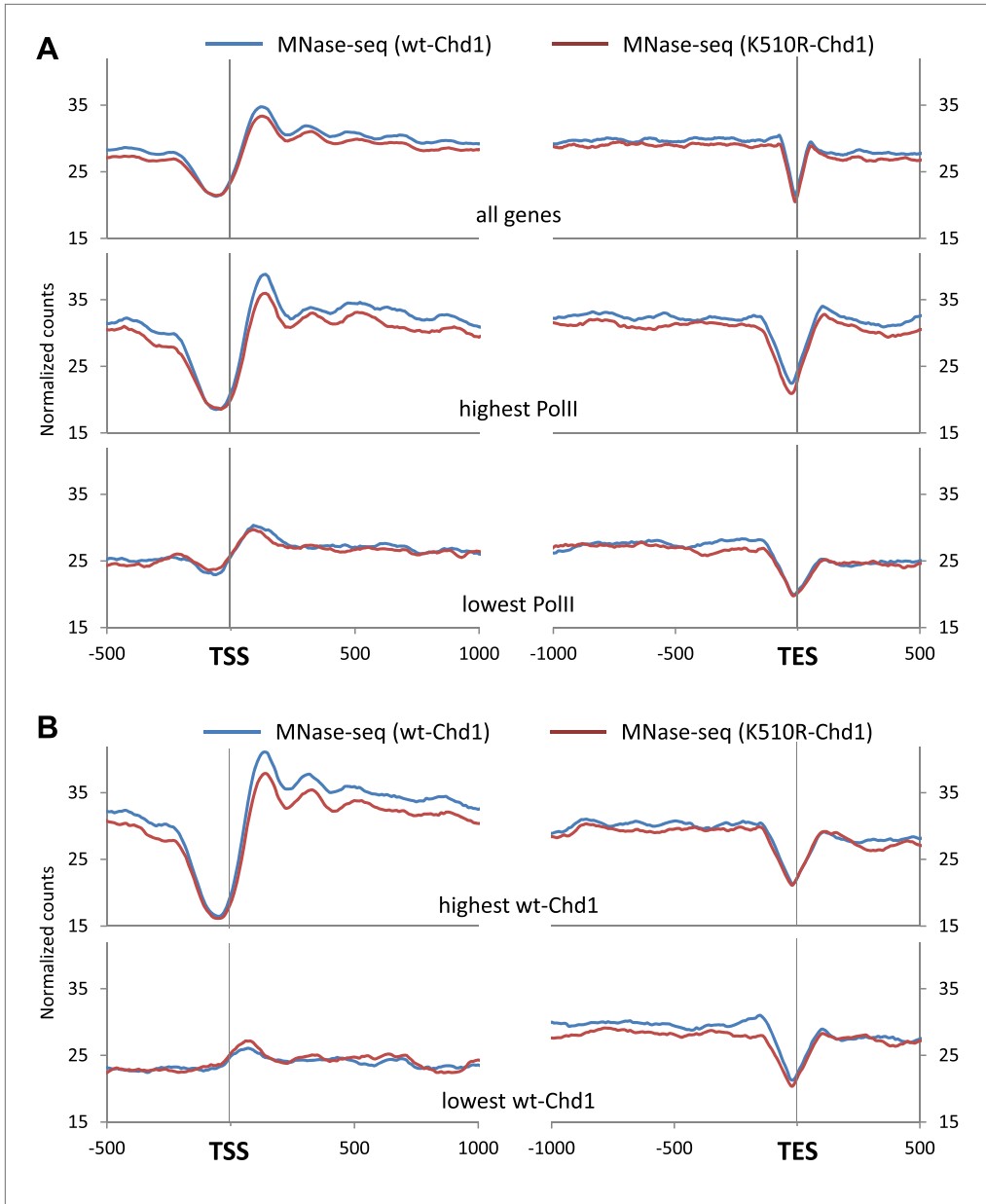

**Figure 2**. Chd1 activity is required to maintain nucleosome occupancy in the promoter region and the gene body. Cross-linked chromatin was digested with MNase and the DNA fragments were subjected to paired-end sequencing. Mono-nucleosomal fragments (111–140 bp) were aligned relative to the TSS or TES. (**A**) Average nucleosome profile for all genes. Genes were also ranked and split into quintiles based on PolII promoter occupancy (all fragment sizes; density within −100 to +300 bp). The nucleosome map for the highest and lowest quintile is shown. (**B**) Nucleosome profiles for genes with either the highest or lowest quintile of wildtype-Chd1 binding (35–75 bp fragment sizes; density within 0 to +1 kb). Statistical significance was determined using the two sample Kolmogorov–Smirnov (KS) test on the average number of normalized counts within 5 kb downstream of the TSS or upstream of TES. All groups were highly statistically significant (p<1 × 10$^{-9}$; TES of genes with the highest Chd1 at the promoter was significant p<3 × 10$^{-4}$) with the exception of the TSS and TES of genes with the lowest PolII density and the TSS of genes with the lowest Chd1 density which showed no significant change.

The following figure supplements are available for figure 2:

**Figure supplement 1**. Chd1 activity is required to maintain nucleosome occupancy in the promoter region and the gene body.

To establish whether the effects of the K510R mutant can be mimicked by reduced levels of Chd1, endogenous levels of Chd1 were reduced using a short-hairpin RNA (*Figure 2—figure supplement 1B*). Nucleosome occupancy was reduced at both the promoter and the gene body upon knockdown of Chd1, as seen with the expression of K510R-Chd1 (*Figure 2—figure supplement 1C*). To determine the concordance between knockdown and expression of the dominant negative, genes were ranked by transgenic wild-type Chd1 occupancy. Genes with the highest Chd1 occupancy were most affected by knockdown, suggesting that knockdown of Chd1 phenocopied the expression of the dominant negative. Overall, loss of Chd1 activity resulted in reduced nucleosome occupancy at both the promoter and the gene body confirming that Chd1 is involved in defining the chromatin landscape.

## Chd1 has opposing roles in regulating nucleosome turnover at the promoter and the gene body

The mapping of MNase-digested mono-nucleosomes provides information regarding steady-state nucleosome occupancy and positioning, but does not yield insight regarding nucleosome dynamics. We used CATCH-IT (covalent attachment of tags to capture histones and identify turnover) to measure nucleosome turnover (*Deal et al., 2010*). CATCH-IT involves the pulse labeling of newly synthesized proteins and the subsequent purification of newly assembled nucleosomes. Our results showed that histone turnover was most rapid at the nucleosomes flanking either side of the promoter and progressively decreased towards the gene body (*Figure 3A*). We observed marked changes in nucleosome turnover upon the expression of K510R-Chd1. First, nucleosome turnover was significantly reduced on either side of the TSS (*Figure 3A*). Second, nucleosome turnover was significantly increased within the gene body, evident from ~500 bp downstream of the TSS and also in the gene body upstream of the TES (*Figure 3A,B*, *Figure 3—figure supplement 1A*). These results suggest opposing roles for Chd1 in regulating nucleosome turnover at the promoter and the gene body. We found that the role of Chd1 in regulating nucleosome turnover at the promoter and the gene body is independent of gene length (*Figure 3—figure supplement 1B*). This is in contrast to yeast, where a previous study indicated Chd1 protects long genes from nucleosome replacement in the gene body (*Radman-Livaja et al., 2012*). In yeast, the magnitude of the effect on nucleosome turnover was approximately equivalent at the promoter and the gene body (*Radman-Livaja et al., 2012*; *Smolle et al., 2012*). However, we find that the primary action of mammalian Chd1 on driving nucleosome turnover is at the promoter, which might suggest a different function of Chd1 in mammals.

We found that nucleosome turnover in the gene body correlated with increasing levels of elongating PolII (*Figure 3C*). This is consistent with several studies suggesting that there is lower retention of the complete octamer at high transcriptional rates than at lower ones (*Kireeva et al., 2002*; *Thiriet and Hayes, 2006*; *Bintu et al., 2011*). Nevertheless, nucleosome turnover was highly variable at low levels of elongating PolII density, indicating that there is significant gene-to-gene variation in the mechanism of PolII transit through nucleosomes at low rates of transcription. Nucleosome turnover in the gene body was increased in cells expressing K510R-Chd1, with the increase approximately constant at all densities of elongating PolII, suggesting an equivalent contribution by Chd1 in the suppression of nucleosome turnover irrespective of transcription rate (*Figure 3C*; lower panel). We sought to further understand the nature of Chd1 action in the gene body by studying potential recruitment mechanisms. As previously discussed, the loss of nucleosome occupancy within the distal regions of the gene body did not scale with Chd1 recruitment to the promoter. This suggests separable Chd1 recruitment to the promoter and the gene body. We found that the level of both wildtype and K510R-Chd1 recruitment to the gene body positively correlated with elongating PolII density within the gene body (*Figure 3D*). Taken together, these observations suggest that occupancy of Chd1 at the promoter and the gene body correlates with the amount of PolII in the respective genomic compartments. The surprising lack of correlation between Chd1 occupancy and suppression of nucleosome turnover in the mutant suggests that Chd1 is not absolutely required for turnover at high transcription rates. Overall, these results suggest that Chd1 is recruited to actively transcribing chromatin where it is involved in the suppression of nucleosome turnover within the gene body.

## Chd1 is responsible for most of PolII-directed nucleosome turnover around promoters

In contrast to nucleosome turnover at the gene body, nucleosome turnover at the promoter was decreased upon the expression of K510R-Chd1. We sought to determine if this reduction correlated

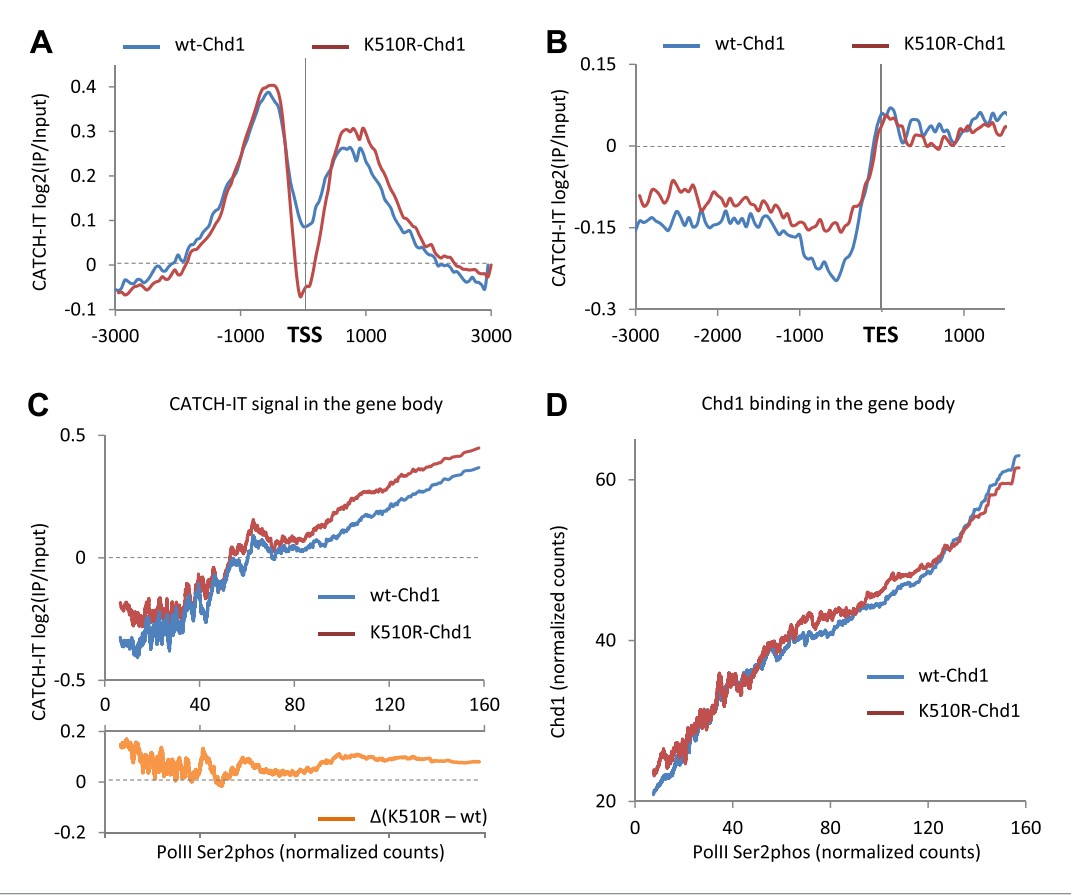

**Figure 3**. Chd1 activity has opposing effects on nucleosome turnover at the promoter and the gene body. Nucleosome turnover is decreased over the promoter but increased over the gene body. The genome-wide average nucleosome turnover was discerned using CATCH-IT at both the (**A**) TSS and (**B**) TES. (**C**) Nucleosome turnover correlates with elongating PolII density and is increased in cells expressing K510R-Chd1. Genes were ranked by the density of serine-2-phosphorylated PolII within the last 3 kb of the gene body. The average density of elongating PolII was calculated with a sliding window of 500 genes plotted against the average CATCH-IT signal within −3 to −0.5 kb relative to the TES (Pearson correlation coefficient >0.85). The difference for each window was calculated and is shown graphically below. (**D**) Chd1 is recruited to genes with actively elongating PolII. Genes were ranked as per (**C**) and plotted against the average ChIP-seq signal for FLAG-tagged wildtype Chd1 (35–75 bp fragment sizes) within the same region. (Pearson correlation coefficient >0.9).

The following figure supplements are available for figure 3:

**Figure supplement 1**. Chd1 activity suppresses nucleosome turnover in the gene body, and the role of Chd1 is independent of gene length.

with PolII promoter density given that both Chd1 occupancy and loss in nucleosome occupancy was most pronounced at genes with high PolII promoter density. Genes with higher levels of PolII displayed increased nucleosome turnover surrounding the TSS (*Figure 4A*), as might have been expected from the positive correlation between steady-state RNA levels and nucleosome turnover (*Deal et al., 2010*; *Yang et al., 2013*). Turnover outside of the promoter region remained high in the absence of Chd1 activity, but was markedly reduced within the promoter proximal region (−350 to +350 bp) encompassing the −1 nucleosome, the TSS and the +1 and +2 nucleosomes (*Figure 4A,B*). We determined the average CATCH-IT signal within the promoter proximal region as a function of PolII occupancy (*Figure 4C*). At low to intermediate PolII densities, we found that cells expressing wildtype-Chd1 displayed a positive correlation between PolII density and nucleosome turnover, with turnover reaching a ceiling value at high levels of PolII. In contrast, expression of the dominant negative

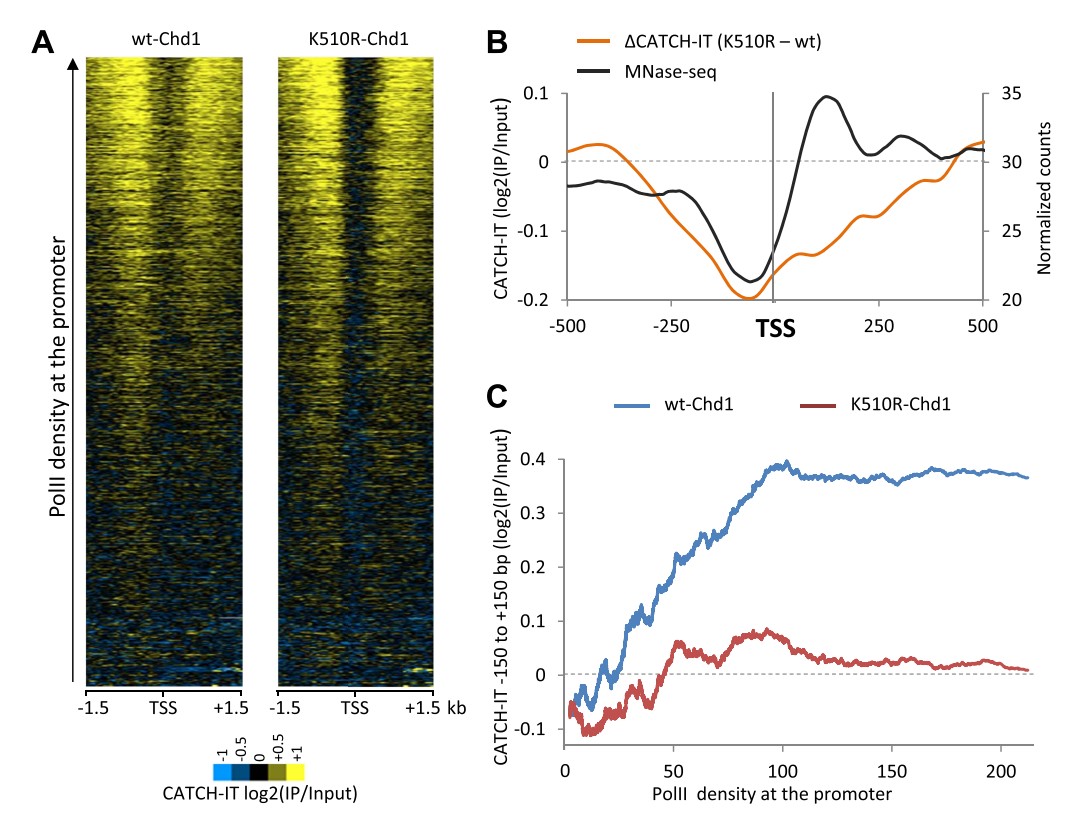

**Figure 4**. Chd1 activity is responsible for PolII-directed nucleosome turnover at the promoter. (**A**) CATCH-IT data represented as a heatmap for the ±1.5 kb surrounding the TSS. Genes were ranked by the level of PolII occupancy at the promoter (all fragment sizes; density within −100 to +300 bp). (**B**) Nucleosome turnover is reduced over the promoter proximal region in cells expressing K510R-Chd1. The difference in CATCH-IT signal between cells expressing wildtype and K510R-Chd1 at the TSS ±500 bp is plotted. The genome-wide average nucleosome occupancy is shown for reference for cells expressing wildtype-Chd1. (**C**) Chd1 is required for PolII-directed turnover at the promoter proximal region. Genes were ranked by PolII promoter density in cells expressing wildtype-Chd1 and plotted against the average CATCH-IT signal in the promoter proximal region with a sliding window of 600 genes.

The following figure supplements are available for figure 4:

**Figure supplement 1**. Knockdown of endogenous Chd1 shows a partial reduction in nucleosome turnover at the promoter.

Chd1 resulted in the near-complete abrogation of turnover to basal levels for all levels of PolII occupancy. Knockdown of endogenous Chd1 resulted in a partial reduction in nucleosome turnover at the promoter (*Figure 4—figure supplement 1*), consistent with either incomplete knockdown or redundant factors. The striking loss of nucleosome turnover in the K510R-Chd1 mutant suggests that Chd1 is responsible for the vast majority of PolII-directed nucleosome turnover within the promoter proximal region. In summary, by analyzing nucleosome dynamics in combination with steady-state nucleosome occupancy, our results suggest that Chd1 is responsible for PolII-directed nucleosome turnover at promoters but suppresses nucleosome turnover within gene bodies. Despite these opposite actions of Chd1 in nucleosome turnover kinetics, we find that Chd1 is required to maintain net nucleosome occupancy both around the promoter and within the gene body.

## Chd1 activity is required for efficient promoter escape by PolII

Previous work in yeast has not identified a clear link between the loss of Chd1 and the consequence on transcription. In fission yeast, approximately half of Chd1 ortholog bound promoters showed a change in nucleosomal occupancy, with the vast majority increasing histone H3 density upon loss of

Chd1 and no significant change in gene expression (*Walfridsson et al., 2007*). In vitro analysis of the *S. cerevisiae PHO5* gene promoter indicated that Chd1 was required to remove the +1 nucleosome, which occludes the TSS, but no robust change in gene expression was observed in vivo without the synergistic loss of the chromatin remodeler ISWI (*Ehrensberger and Kornberg, 2011*). To better interpret the transcriptional consequences of the alterations in mammalian chromatin architecture at the promoter, we established the changes in PolII distribution upon K510R-Chd1 expression by performing ChIP-seq with an antibody against the N-terminus of the largest subunit of PolII, which recognizes total PolII independent of phosphorylation (*Figure 5A*). The majority of PolII was bound to the promoter region, as expected given the prevalence of promoter proximal pausing in metazoans and consistent with the idea that promoter escape is a major barrier to productive elongation (*Krumm et al., 1995*; *Rahl et al., 2010*). This is in contrast to *S. cerevisiae*, where the majority of transcriptional control is at the level of PolII recruitment to the promoter, with a relatively even distribution of PolII across the gene (*Stargell and Struhl, 1996*; *Robert et al., 2004*; *Rhee and Pugh, 2012*). Unexpectedly, the expression of K510R-Chd1 resulted in increased PolII occupancy within the promoter region and was most apparent at genes with the highest PolII occupancy, suggesting a role for Chd1 in the distribution of PolII.

The increased PolII occupancy at the promoter could be due to either increased recruitment of PolII to the promoter or alternatively decreased efficiency of promoter escape into the gene body. To distinguish these alternatives, we established the distribution of actively elongating PolII using an antibody that recognizes the serine-2-phosphorylated elongating form (PolII Ser2phos) (*Figure 5B*). The signal for PolII Ser2phos increased after the promoter region and continued into the gene body, as expected for elongating PolII. In cells expressing K510R-Chd1, the level of PolII Ser2phos was reduced throughout the gene body, with the effect most pronounced at the genes with highest level of PolII occupancy at the promoter, indicating a role of Chd1 in enabling promoter escape. Alternatively, Chd1 might be affecting the elongation rate within the gene body. To test this possibility, the distribution of PolII Ser2phos was normalized to the average density across the first 5 kb of the gene body (*Figure 5C*). We found no significant change in the distribution of elongating PolII, suggesting that Chd1 does not affect elongation rate in the gene body. Altogether, our results showed that Chd1 activity is required for efficient promoter escape by PolII and in its absence, PolII accumulates at the promoter with a concomitant decrease in the gene body.

## Chd1 alleviates the nucleosome barrier to PolII transit at processive genes

As Chd1 modulates PolII release from promoters, we wondered whether this ATP-remodeling complex alleviates PolII stalling. Therefore, we calculated the stalling index in the presence of wild-type and mutant Chd1. We defined the stalling index as the ratio of total PolII density at the promoter (−100 to +300 bp) relative to the density of PolII Ser2phos within the gene body (+1 to +5 kb) (*Figure 6A*, *Figure 6—figure supplement 1A*). We found that, on average, cells expressing K510R-Chd1 had a two-fold increase in the stalling index in comparison to wildtype (*Figure 6B*). To determine if this increase in stalling predominantly occurred at stalled or processive genes, genes were ranked by increasing stalling index (*Figure 6C*). Cells expressing wild-type Chd1 displayed stalling indices spanning several orders of magnitude. Cells expressing K510R-Chd1 had a greatly reduced range of stalling indices, with an overall increase in stalling and the processive genes being most affected. This suggests that Chd1 enables PolII to escape from the promoter region to allow processive transcription.

To confirm that the main function of Chd1 is at processive genes, Chd1 binding to stalled or processive genes was addressed. As Chd1 binding scales with PolII density (*Figure 1*), only the genes in the highest quintile of PolII promoter occupancy were investigated to minimize the variation due to differences in PolII promoter occupancy. We found that processive genes showed a much higher Chd1 occupancy than stalled genes (*Figure 6D*), despite having a lower level of PolII density (*Figure 6—figure supplement 1B*). In addition, the median stalling index was increased ~twofold at these processive genes by the expression of K510R-Chd1, whereas these stalled genes were unaffected (*Figure 6—figure supplement 1C*). This suggests that although Chd1 recruitment largely follows PolII occupancy at the promoter, there must be additional factors that are linked to processive elongation that influence Chd1 recruitment.

The Chd1-binding profile was plotted for processive genes over the promoter region to understand how Chd1 might be functioning to enable promoter clearance at processive genes (*Figure 6E*). Wildtype-Chd1 displayed a relatively even profile across the promoter, whereas K510R-Chd1 displayed

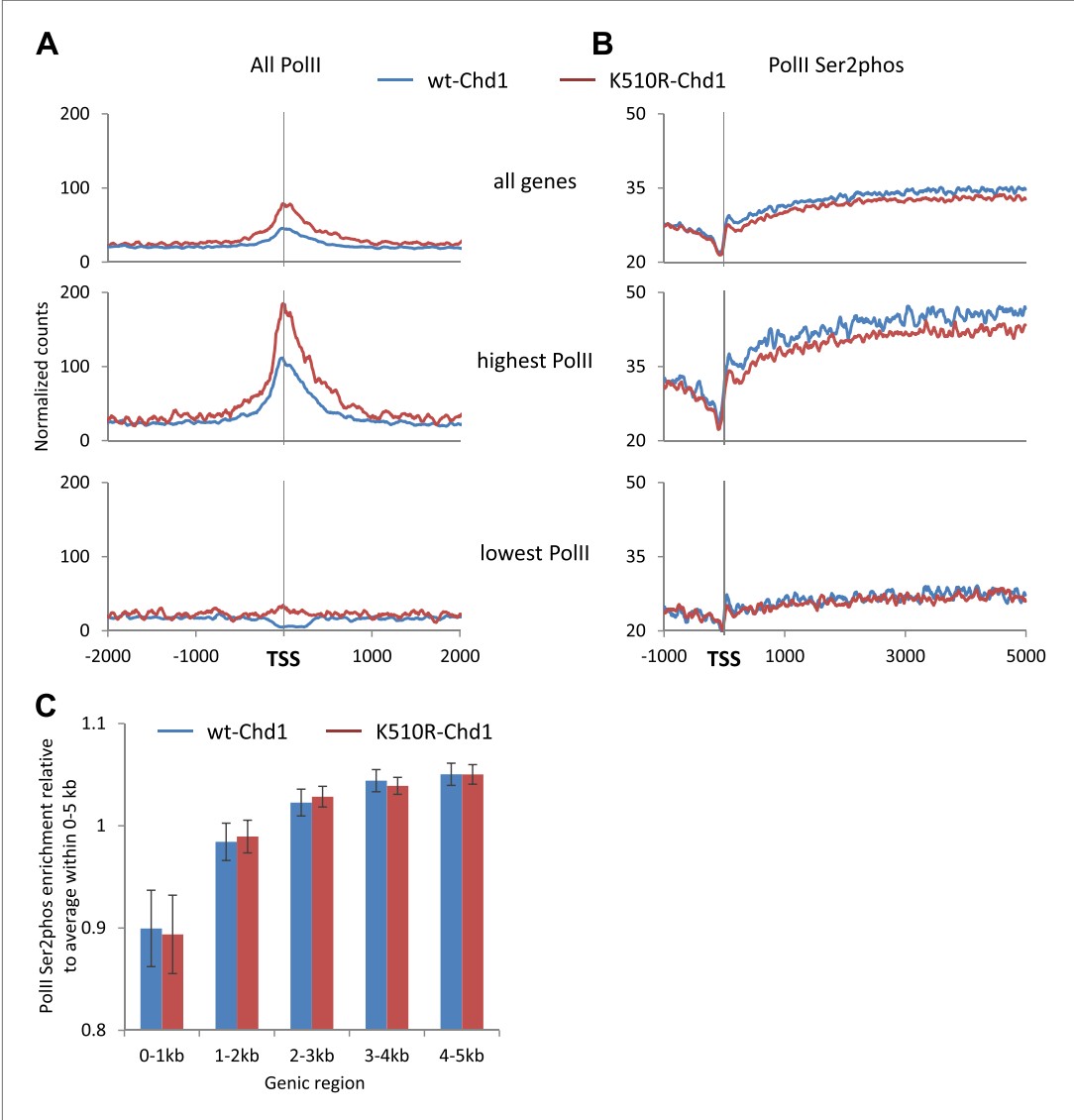

**Figure 5**. Chd1 activity is required to enable PolII to efficiently escape the promoter proximal region. Genome-wide distribution of PolII was determined using ChIP-seq with antibodies against (**A**) the N-terminus of Rpb1 to map total PolII and (**B**) PolII Ser2phos to map elongating PolII. The recovered short reads (35–75 bp) were mapped and the genome-wide average was plotted. Genes were also ranked and split into quintiles based on total PolII promoter occupancy in cells expressing wildtype-Chd1 (all fragment sizes; density within −100 to +300 bp). The PolII distribution is shown for the highest and lowest quintiles. Statistical significance was determined using the KS test on the average counts within (**A**) −100 to +300 bp and (**B**) +1 kb to +5 kb relative to the TSS. Differences between wt-Chd1 and K510R-Chd1 were significant ($p < 1 \times 10^{-9}$) with the exception of elongating PolII density in genes the lowest promoter PolII occupancy. (**C**) The elongation rate within the gene body is unaffected by the expression of K510R-Chd1. The fold enrichment in PolII Ser2phos density within 1 kb genic windows was calculated relative to the average density within the first 5 kb of the gene. Error bars represent standard deviation.

peaks of binding near the dyad axis for each of the promoter proximal nucleosomes, with the peak of Chd1 ~50 bp inside of the leading edge of the +1 nucleosome. Binding at this position is consistent with in vitro studies, which suggest that chromatin remodelers initiate translocation on nucleosomal DNA at an internal position ~20 bp from the dyad at superhelical location 2 (*Saha et al., 2005*; *Zofall et al., 2006*). These peaks likely represent the sites of action of wildtype-Chd1 on its nucleosomal substrate, but through the expression of catalytically dead Chd1 we have been able to capture these normally transient interactions.

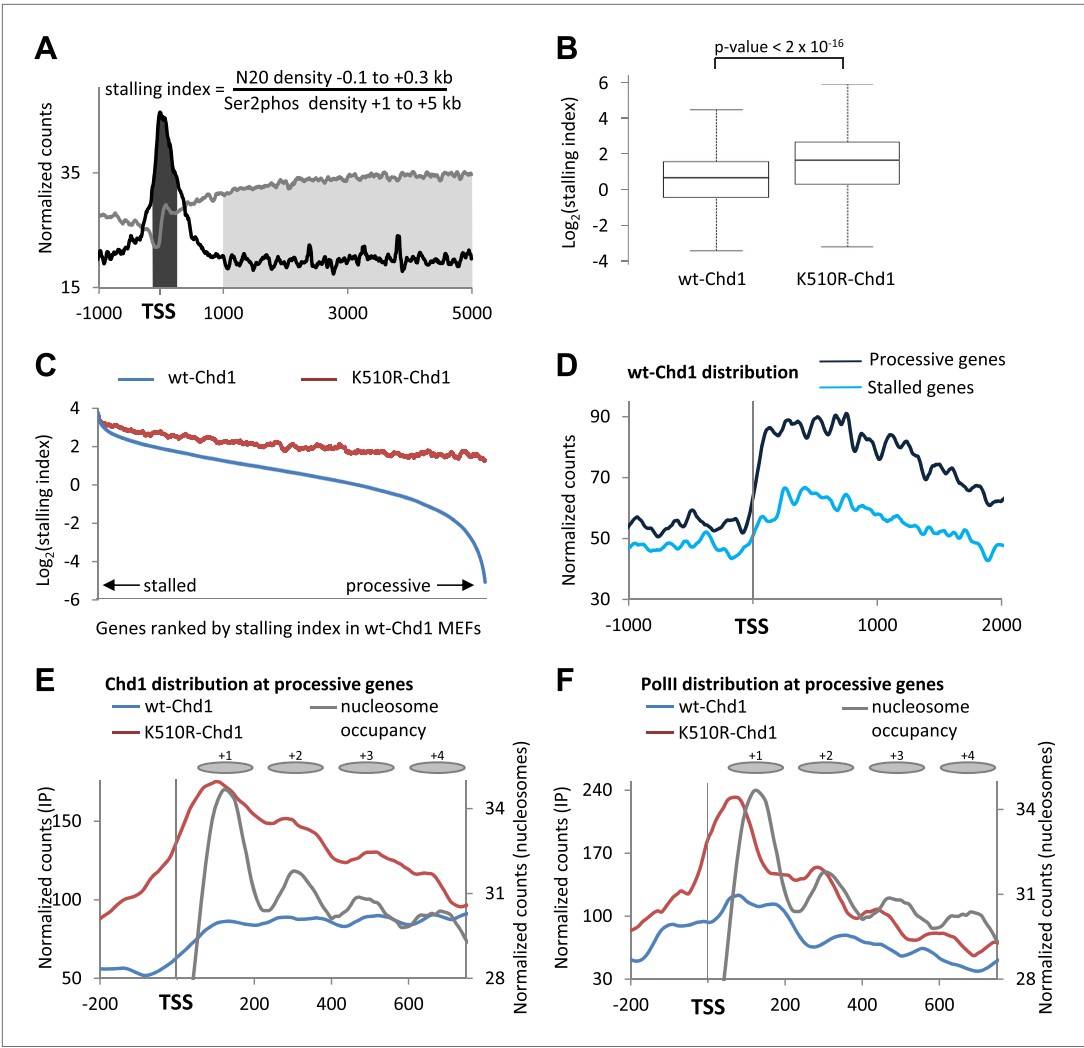

**Figure 6**. Chd1 functions to alleviate the nucleosomal barrier to PolII transit at highly processive genes. (**A**) Schematic describing the calculation of the stalling index. The dark gray shading indicates the total PolII density at the promoter (−100 to +300 bp) as calculated from the ChIP-seq data using the N20 antibody against the N-terminus of Rpb1 (black line). The light grey shading indicates the density of actively elongating PolII in the gene body (+1 to +5 kb or the end of the gene), calculated from the PolII Ser2phos ChIP-seq data (gray line). The stalling index is the ratio of these two values. (**B**) The stalling index is increased ~twofold in cells expressing K510R-Chd1. Box plot comparing the distribution of stalling indices for all genes. Statistical significance was determined using the Kolmogorov–Smirnov test. (**C**) Lack of Chd1 leads to increased PolII stalling, preferentially affecting genes with processive transcription. Genes were ranked by stalling index and plotted against the stalling index in cells expressing wildtype or K510R Chd1 with a sliding window of 200 genes. (**D**) Chd1 is preferentially recruited to the promoter region of genes with processive transcription compared to stalled genes. Wildtype-Chd1 binding profile across highly stalled and processive genes from the highest quintile of PolII promoter occupancy. (**E**) K510R-Chd1 accumulates near the dyad axis of the promoter proximal nucleosomes. The wildtype and K510R Chd1 binding profiles at these highly processive genes are shown. Nucleosome occupancy in cells expressing wildtype-Chd1 is shown for reference with gray ovals indicating the 147 bp of DNA protected by the nucleosomes. (**F**) PolII stalls in front of promoter proximal nucleosomes in the absence of Chd1 activity. PolII distribution at highly processive genes, as measured by ChIP-seq using the N20 antibody is shown. Nucleosome occupancy in cells expressing wildtype-Chd1 is shown for reference with gray ovals indicating the 147 bp of DNA protected by the nucleosomes.

The following figure supplements are available for figure 6:

**Figure supplement 1**. Loss of Chd1 results in an increase in the stalling index, predominantly at processive genes.

*Figure 6. Continued on next page*

*Figure 6. Continued*

**Figure supplement 2**. The promoters of processive genes have a highly dynamic chromatin barrier to transcription.

Recent work in *Drosophila* indicated that at the more processive M1BP bound genes, PolII accumulated in front of the well-positioned +1 nucleosome (*Li and Gilmour, 2013*). In agreement with this, we find that processive genes have higher promoter proximal nucleosome occupancy and furthermore that +1 nucleosome turnover is higher at processive genes than stalled genes (*Figure 6—figure supplement 2A,B*). This raised the possibility that the promoter proximal nucleosomes at processive genes are acting as dynamic barriers to transcription. We therefore sought to understand how PolII transits through the promoter proximal nucleosomes in the absence of functioning Chd1. The total PolII ChIP-seq profile was plotted over the promoter region of these highly processive genes, which become stalled following the expression of K510R-Chd1. In contrast to the relatively even profile for PolII in the cells expressing wildtype-Chd1, the absence of Chd1 activity resulted in clear peaks in the PolII distribution (*Figure 6F*). These peaks precede each of the promoter proximal nucleosomes and are shifted upstream relative to the sites of action of Chd1 at the nucleosomes, with PolII accumulating ~20 bp inside the entry site of the +1 nucleosome. This suggests that Chd1 functions at processive genes to remove the nucleosomal barrier ahead of PolII and in its absence PolII stalls in front of the now static promoter proximal nucleosomes leading to a reduced efficiency of promoter escape.

## Discussion

Using a novel high resolution chromatin immunoprecipitation strategy, we report the key role of Chd1 in defining nucleosome dynamics during the transcription cycle (*Figure 7A*). We find that mammalian Chd1 is primarily recruited just downstream of the start site of PolII-bound genes. Through the expression of a dominant negative form of the chromatin remodeler Chd1, we have shown that Chd1 is responsible for the vast majority of nucleosome turnover at the promoter and is required for PolII promoter escape. Our analysis identifies promoter proximal nucleosomes as a barrier to PolII promoter escape and suggests how Chd1 overcomes this barrier. In addition, we find that Chd1 also functions in later stages of the transcription cycle, with Chd1 binding in the gene body correlating with elongating PolII density, where it suppresses nucleosome turnover and in its absence leads to decreased nucleosome occupancy.

The role for Chd1 in gene bodies is consistent with in vitro studies indicating that Chd1 actively assembles chromatin and slides nucleosomes into ordered arrays (*Lusser et al., 2005*; *Torigoe et al., 2013*). In vivo, Chd1 is likely to function in reassembly of nucleosomes displaced during elongation. In the absence of Chd1 activity, reassembly is compromised, leading to the deposition of de novo synthesized nucleosomes, consistent with observations in yeast (*Gkikopoulos et al., 2011*; *Hennig et al., 2012*; *Radman-Livaja et al., 2012*; *Smolle et al., 2012*). The close correlation between Chd1 occupancy and elongating PolII density indicates a possible mechanism for the recruitment of Chd1, consistent with studies indicating that Chd1 associates with elongation factors including polymerase-associated factor (PAF1), DSIF and FACT (*Simic et al., 2003*). In vitro remodeling by Chd1 is dependent upon its C-terminal DNA binding domain interacting with extranucleosomal DNA and may also be required for efficient Chd1 recruitment (*Kagalwala et al., 2004*; *Stockdale et al., 2006*; *McKnight et al., 2011*). More specifically, the DNA immediately upstream of the nucleosome has been shown to be important for the initial stage of nucleosome movement in ISWI-type remodelers (*Zofall et al., 2004*). Single-molecule optical trap experiments suggest that PolII transcribes through a nucleosome by the *in cis* transfer of histones, particularly the H3:H4 tetramer, to DNA upstream of PolII via the formation of a transient DNA loop (*Hodges et al., 2009*; *Kulaeva et al., 2009*; *Bintu et al., 2011*). This might suggest that Chd1 is recruited to the gene body via the elongation complex and is then targeted to the nucleosome substrate via binding to the exposed DNA loop immediately adjacent to the histone octamer to facilitate the *in cis* transfer of histone octamers.

Chd1 is primarily found at the 5′ end of PolII bound genes and is particularly enriched at genes undergoing processive transcription. By increasing the resolution of standard cross-linked ChIP-seq, we find that Chd1 binds within the leading edge of promoter proximal nucleosomes, where it is required for virtually all of PolII-directed nucleosome turnover at the promoter. Expression of

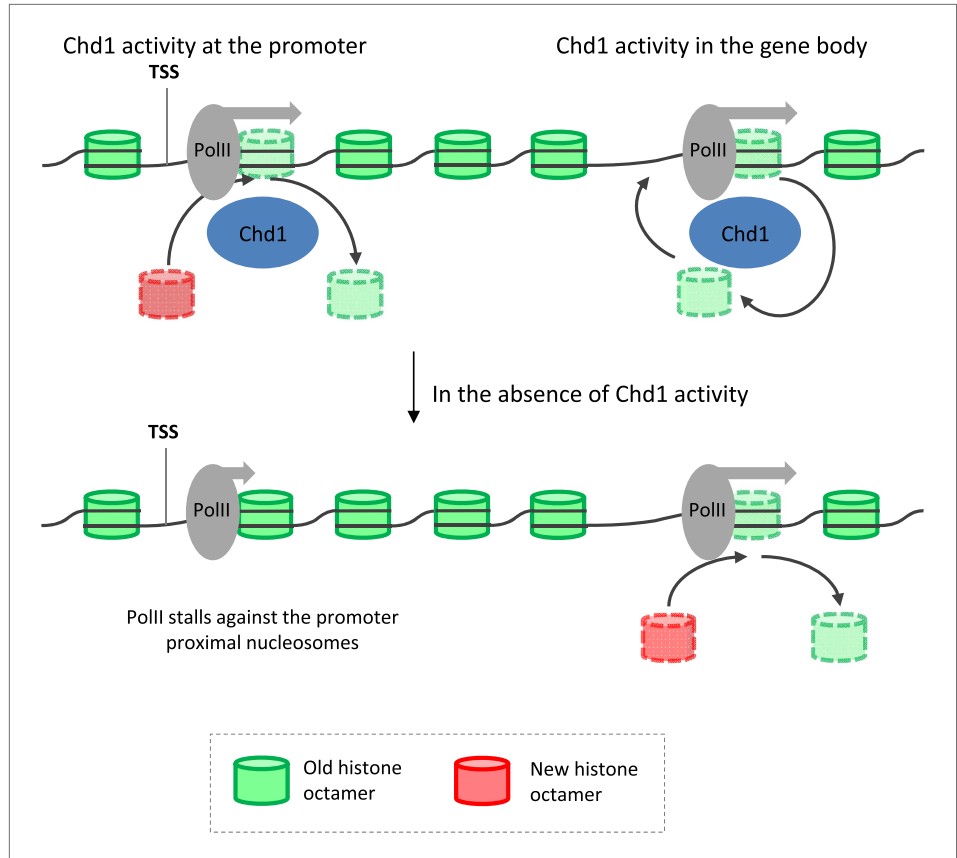

**Figure 7**. Model depicting the role of Chd1 in defining nucleosome dynamics during transcription. At the promoter of processive genes, Chd1 is recruited directly to the promoter proximal nucleosomes, where it is required for both eviction of existing histone octamers (green) and deposition of new octamers (red). In the absence of Chd1 activity, PolII stalls close to the leading edge of the nucleosome. In the gene body, Chd1 occupancy correlates with elongating PolII, where it reassembles histone octamers in cis behind the transiting PolII. The lack of Chd1 activity leads to increased deposition of new histone octamers through an unknown factor.

K510R-Chd1 results in reduced steady-state nucleosome occupancy. Despite this, PolII was observed to accumulate just within the entry site of promoter proximal nucleosomes and a concomitant increase in the stalling index, indicative of an increase in the barrier to PolII promoter escape. This increased PolII stalling is consistent with the reduced nucleosome turnover at the promoter, indicating a more static chromatin structure forming a barrier to promoter escape. The increase in stalling occurs despite a decrease in steady-state nucleosome occupancy, highlighting the importance of kinetic measurements for chromatin structure. The apparent discrepancy of an increase in the barrier to PolII accompanying a decrease in steady-state nucleosomal occupancy can be most easily reconciled if Chd1 acts to evict the nucleosomes ahead of PolII transit and then reassemble nucleosomes in the wake of PolII. In the absence of Chd1, nucleosome eviction is likely to still occur as a result of transcription albeit at a lower rate, and the CATCH-IT analysis suggests de novo nucleosome deposition is greatly reduced, overall leading to the reduced steady-state nucleosome occupancy.

These opposing activities at the gene body and the promoter may perhaps be explained through differing mechanisms by which Chd1 engages with the nucleosomal substrate after the point of recruitment. As discussed, the assembly activity in the gene body is likely to be directed via the non-specific DNA binding domain interacting with extra-nucleosomal DNA. A recent in vitro study directly tethered yeast Chd1 to the nucleosomal substrate independent of the DNA binding domain and observed nucleosome sliding and eviction more typical of SWI/SNF-type remodelers (*Patel et al., 2013*). This suggests that the observed nucleosome eviction by Chd1 at the promoter is likely to reflect activity independent of the DNA binding domain, raising the question of how Chd1 engages with the

nucleosomal substrate. The tandem chromodomains in human Chd1 have been shown to specifically interact with H3K4me3, whereas in *S. cerevisiae*, the chromodomains lack key tryptophan residues required for H3K4me3 binding (*Flanagan et al., 2005*; *Sims et al., 2005*). Therefore activity dependent on the chromodomains provides a tantalizing explanation for the role of mammalian Chd1 in nucleosome eviction at the promoter in contrast to yeast Chd1, where Chd1 does not affect occupancy of the +1 nucleosome (*Gkikopoulos et al., 2011*). This unique requirement for mammalian Chd1 to overcome the nucleosomal barrier to PolII at the promoter may reflect differences from yeast in both chromatin architecture and transcriptional regulation, with yeast having the TSS buried within the +1 nucleosome and primarily regulating transcription through recruitment (*Robert et al., 2004*; *Rhee and Pugh, 2012*).

These functions of mammalian Chd1 in defining nucleosome dynamics during transcription may explain phenomena associated with its loss. Given that the majority of post-translational modifications (PTMs) are found on either H3 or H4, the retention of the H3:H4 tetramer during transcription has been proposed to have a role in the maintenance of chromatin states throughout the genome (*Kulaeva et al., 2009*). This suggests that the increased histone turnover observed in the gene body in the absence of Chd1 activity could compromise epigenetic regulation, and might explain the observation that Chd1 is required for maintaining euchromatin in ES cells by preventing heterochromatin formation (*Gaspar-Maia et al., 2009*). In contrast, the Chd1 dependent turnover observed at the promoter would have the effect of removing histone 'memory' and might explain the requirement for Chd1 in reprogramming through the deposition of de novo histones at the promoter to allow the reactivation of pluripotency genes (*Gaspar-Maia et al., 2009*; *Jullien et al., 2012*; *Skene and Henikoff, 2012*). Given that doxorubicin, a commonly used anti-cancer drug, enhances nucleosome turnover, the identification of Chd1 in directing nucleosome turnover provides a mechanistic basis for its tumor suppressor role and may indicate the molecular basis for carcinogenesis in the absence of Chd1 (*Huang et al., 2012*; *Liu et al., 2012*; *Yang et al., 2013*). Overall, these observations point to a key role for Chd1 in regulating the mammalian transcriptional landscape.

## Materials and methods

### Constructs and cell lines

Immortalized MEFs were transfected with a *Piggybac* expression vector with the CAG promoter driving N-terminal FLAG-tagged expression of Chd1 (40086932; Open Biosystems, Pittsburgh PA) and puromycin resistance. The K510R mutation was generated by site directed mutagenesis and verified by sequencing. MEFs were cultured in Dulbecco's modified Eagle's medium (DMEM) (#11965-092; Invitrogen, Grand Island NY) supplemented with 10% FBS, 1% penicillin-streptomycin at 37°C and puromycin (1.5 µg/ml). Knockdown of endogenous Chd1 was achieved using vectors as previously described (*Gaspar-Maia et al., 2009*).

### Western blots

Western blots were performed under standard conditions using whole cell extracts and probed with the following antibodies: FLAG (M2 F1804; Sigma, St. Louis MO); Chd1 (39729; Active Motif, Carlsbad CA); γ-Tubulin (ab11316; Abcam, Cambridge MA).

### ChIP-seq

ChIP was performed as previously described (*Schmiedeberg et al., 2009*) with the following modifications. Chromatin fragmentation was achieved by adding MNase after cell lysis to produce predominantly mono-nucleosomes. Sonication (40 s; 30% amplitude; Branson digital sonifier) was used to ensure complete extraction of Chd1 and histone H3, as measured by western analysis. The following antibodies were used: FLAG (A2220; Sigma); N20 (sc-899; Santa Cruz, Dallas Tx); PolII Ser2phos (ab5095; Abcam). Libraries were prepared from the isolated DNA and sequenced on the Illumina HiSeq 2000 platform (*Henikoff et al., 2011*). Paired-end sequencing data were processed and aligned to the mm9 genome build using Bowtie and the distribution plotted around the 5′ and 3′ ends of genes as previously described (*Henikoff et al., 2011*).

### CATCH-IT

CATCH-IT was performed as described previously with the following modifications (*Yang et al., 2013*). Nuclei were isolated by hypotonic lysis of cells in ice-cold NE1 buffer (20 mM HEPES-KOH (pH 7.9), 10 mM KCl, 1 mM $MgCl_2$, 0.1% Triton X-100, 1 mM DTT, 20% Glycerol) and then processed as previously described.

Input and CATCH-IT DNAs were amplified using a whole-genome amplification kit (Sigma-Aldrich), followed by labeling using Cy3 and Cy5 heptamers according to the Roche Nimblegen labeling protocol. Labeled samples were hybridized together to mouse 2.1-million probe promoter arrays (Roche Nimblegen, Madison WI).

## Accession numbers

Data have been submitted to the Gene Expression Omnibus under accession number GSE52349 (*Skene et al., 2014*).

## Acknowledgements

We thank Jorja Henikoff for bioinformatic support. PJ Skene is a Damon Runyon Fellow supported by the Damon Runyon Cancer Research Foundation (DRG-2110-12). This work was supported by the Howard Hughes Medical Institute.

## Additional information

### Funding

| Funder | Grant reference number | Author |
| --- | --- | --- |
| Damon Runyon Cancer Research Foundation | DRG-2110-12 | Peter J Skene |
| Howard Hughes Medical Institute | | Aaron E Hernandez, Steven Henikoff |
| National Institutes of Health | R01 HL65440 | Mark Groudine |
| National Institutes of Health | U54 CA143862, R01 ES020116 | Steven Henikoff |

The funders had no role in study design, data collection and interpretation, or the decision to submit the work for publication.

### Author contributions

PJS, Conception and design, Acquisition of data, Analysis and interpretation of data, Drafting or revising the article; AEH, Aiding in generating Chd1 expression constructs; MG, Conceptual design, Editing the article; SH, Conception and design, Drafting or revising the article

## Additional files

### Major dataset

The following dataset was generated:

| Author(s) | Year | Dataset title | Dataset ID and/or URL | Database, license, and accessibility information |
| --- | --- | --- | --- | --- |
| Skene PJ, Hernandez AE, Groudine M, Henikoff S | 2013 | The nucleosomal barrier to promoter escape by RNA Polymerase II is overcome by the chromatin remodeler Chd1 | GSE52349; http://www.ncbi.nlm.nih.gov/geo/query/acc.cgi?acc=GSE52349 | Publicly available at the Gene Expression Omnibus (http://www.ncbi.nlm.nih.gov/geo/). |

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
