## [Decision Letter]

[Editors’ note: although it is not typical of the review process at *eLife*, in this case the editors decided to include the reviews in their entirety for the authors’ consideration as they prepared their revised submission.]

Thank you for sending your work entitled “The nucleosomal barrier to promoter escape by RNA Polymerase II is overcome by the chromatin remodeler Chd1” for consideration at *eLife*. Your article has been evaluated by a Senior editor and 3 reviewers, one of whom is a member of our Board of Reviewing Editors.

As you will gather from the full reviews included below, there was a clear disparity in the initial assessment of your manuscript. Whereas Reviewers 1 and 2 observed the various strengths of your work and found it appropriate for publication in *eLife,* Reviewer 3 pointed out that several key observations into the role of Chd1 in nucleosome remodeling at distinct genomic locations described in your manuscript had been previously established in the literature. This led Reviewers 1 and 2 to re-review your manuscript to determine to what degree the novel aspects in your work constituted a significant advance in the field. After this process, Reviewers 1 and 2 agreed that overall the paper ranks as very good and that there is new information in the manuscript.

Upon further discussion, we would like to encourage a resubmission of a revised manuscript that not only thoroughly addresses the technical and data interpretation concerns raised by Reviewer 3, but also is rewritten in a way that better emphasizes what was known and what is novel in this paper about the role of Chd1 in nucleosome position and exchange. We also encourage addition of any new available data that would further advance our understanding of the molecular mechanism of action of Chd1.

*Reviewer*
*1:*

The manuscript by Skene et al clearly demonstrates a key role for the nucleosome remodeler Chd1 in RNA polymerase II (RNAPII)-driven nucleosome turnover at gene promoters in vertebrate cells. Accordingly, Chd1 arises as a key regulator of RNAPII activity at the level of pausing and elongation. Given the widespread occurrence of RNAPII pausing in metazoans and the ever-increasing appreciation of RNAPII elongation control in regulation of gene expression, the current manuscript is deemed of high impact for the gene expression and chromatin fields.

The manuscript is beautifully written, the data is of high quality and generated with cutting-edge technology and the conclusions are fully supported by data. In my mind, the manuscript is meritorious for publication in *eLife* basically as it is.

My only minor concern is the creation of the RNAPII 'stalling index' in Figure 6, which is basically equivalent to the RNAPII 'pausing index' used by many reports in the literature. There is a minor technical difference in how Skene et al. calculated their index (i.e., using Ser2-phospho-RNAPII as denominator instead of total RNAPII) which I don't mind, but I don't think there is need to introduce 'stalling' here, I'll be happy with 'pausing'.

*Reviewer*
*2:*

The authors are experts in chromatin biology, and have developed several assays that enable different attributes of promoter chromatin to be analyzed. Here, they explore the impact of Chd1 remodeler on promoter chromatin. This issue has been addressed previously in ES cells at lower resolution, where promoter association was observed, with Chd1 loss leading to the acquisition of heterochromatin.

The paper starts with the application of methods developed in the Henikoff lab to examine both Chd1 and RNA PolII in MEFs. This involves the separate examination of larger and smaller fragments derived from limited MNase digestion of crosslinked chromatin, following chromatin IP (ChIP). The larger fragments represent linkage of the factor to nucleosomes, whereas the smaller represent the factor's footprint. Support is provided that smaller fragments with PolII better link to the TSS.

For Chd1, the authors examine both an epitope-tagged WT and a catalytic mutant. With both Chd1 derivatives, localization to the +1 nuc was observed, however the catalytic mutant showed additional occupancy of proximal downstream nucleosomes. This data was nicely done and described. The authors then explored the impact on nucleosome occupancy and positioning following expression of the dominant-negative Chd1 derivative. Here, they claim a loss in nucleosome occupancy without a change in positioning. I find the effect very small overall, even when PolII levels are considered, but much more clear in the case where the highest quintile of Chd1 occupancy is examined. The authors note a difference in impact here in MEFs on the +1 nuc, vs in budding yeast where no change is observed following Chd1 loss, which I found interesting. I also found the correlation of Chd1 with 'open' promoters interesting. The authors then apply an alternative shRNA approach to knock down Chd1, and observe similar (and even somewhat stronger) results.

As prior approaches provide steady-state measurements, the authors then apply the CATCH-IT technique to explore dynamics. First, they show that turnover is high at locations flanking the TSS. The authors then explore the impact of expression of the dominant negative. The authors emphasize a significant difference at the locations flanking the TSS, and in the gene body. However, this is really confined to the nucleosome directly on, or adjacent to, the TSS; here it is clear and interesting, however, regarding the gene body, I found the difference with the dominant negative apparent, but the scale/magnitude quite small.

The next section shows the promoter/TSS affect with the dominant negative scales with PolII to distinguish between increased recruitment vs lack of escape, relative distribution of elongating PolII, and (with additional work) showed support for a role in promoter escape. Stalling index analysis show increases with the catalytic mutant and a preference for processive genes, which correlated with wt Chd1 occupancy. I found this data quite interesting. I found the data showing the peak of dominant negative Chd1 occupancy 50 bp inside the nucleosome very interesting, and consistent with current models of DNA translocation. Likewise, the peaks of PolII preceding nucleosomes in the early gene body supports the authors interesting claim of utilizing Chd1 activity to help facilitate PolII passage and reassembly behind the polymerase.

Overall, I found the work well designed, executed and interpreted, and I strongly support its publication.

*Reviewer*
*3:*

This manuscript investigates the role of Chd1 in transcription. Using MNase-seq, PolII ChIP-seq and CATCH-IT in MEF cells expressing WT or a dominant negative form of Chd1, the authors conclude that Chd1 is a factor required for the RNAP to overcome the barrier of the +1 nucleosome and relieve stalling of RNAP, whereas in the gene body it suppresses nucleosome turnover. There are some potentially good ideas in this manuscript, but the data do not significantly extend what we know from previous work and the data suffers from insufficiently rigorous analysis.

These results are similar to work from a number of labs (Workman, Tamkun, Hartzog, Rando, Kornberg), showing that Chd1 localizes at promoters and within gene bodies where it facilitates transcription by altering nucleosome occupancy and turnover/histone replacement, especially at highly active genes. In particular, the data is reminiscent of Radman-Livaja et al. (PLOS Genetics 2012), where it is shown in yeast that loss of Chd1 activity leads to slower turnover of nucleosomes over the promoter and faster turnover in gene bodies and near 3' ends.

The authors emphasize the idea that Chd1 does not remodel promoter nucleosomes in yeast, but I don't understand why (they cite a Science paper from the Owen-Hughes lab that focused on nucleosome positioning). In fact that paper showed lower nucleosome occupancy near promoters in remodeling deficient cells, in total agreement with the current manuscript. Furthermore, the Kornberg lab (Ehrensberger, PNAS 2011) has done nice work in *S. cerevisiae* showing that Chd1 is a promoter-specific chromatin remodeler, and work in *S. pombe* (Walfridsson et al, EMBO 2007) showed that Chd1 localized particularly to promoter regions where it helped remove nucleosomes near the transcription start site to enable transcription. Thus, I don't see significant novelty in the majority of conclusions presented and don't find compelling the suggested connection between Chd1 activity at promoters and the metazoan-specific phenomenon of promoter stalling of RNAP. On this point, the hypothesis that PolII is being blocked in Chd1 mutant cells by nucleosomes despite the overall occupancy of the +1 nucleosome being lower is not clear to me. I can see how PolII could be blocked by an increase in nucleosome occupancy or by more static nucleosomes, but i.e. not what is indicated by this data. What the authors show is reduced +1 occupancy and reduced +1 turnover, which is most simply explained by nucleosomes being removed and not efficiently replaced. There is no clear data that sheds light on how this lower occupancy or failure to re-assemble nucleosomes might inhibit PolII.

Major concerns:

1) There are a number of cases where strong conclusions are drawn based on non-quantitative analysis of normalized group–average profiles. Group averages can be dramatically shifted by a small subset of genes and are not a reliable way to draw generalizable conclusions about a population. In cases where the authors wish to say that Chd1 mutation 'increases' or 'decreases' signal, statistical evaluations should be performed and described clearly in the methods section. When the authors wish to say that two signals 'correlate' with each other, correlation analysis should be performed and Pearson/Spearman R squared or similar values should be presented.

2) It is really hard to interpret the data on PolII ChIP in cells expressing the catalytically dead Chd1 mutant without knowing what the effects of blocking Chd1 activity are on gene expression. The authors should at the very least select some example genes where loss of Chd1 increases the PolII stalling index and measure the nascent RNA levels at that gene by run-on or RT-PCR of intronic regions. The interpretation of the authors is that PolII is being 'blocked' at the promoter by the presence of more stable nucleosomes, but without showing that there is actually less active elongation within the gene bodies, this conclusion is very weakly supported, especially in light of work in *Drosophila* that shows normal Ser2-P PolII (active elongation form) levels in Chd1 mutants (Armstrong and Tamkun labs).

3) It is very surprising that the authors don't mention any of the considerable work done on the relationship between PolII and the +1 nucleosome in *Drosophila*. In particular, Dave Gilmour's recent EMBO paper clearly shows that there are different classes of genes that differ in the role of the +1 nucleosome. Genes with the most paused PolII lack a significant +1 nucleosome signal and are paused by separate factors (GAGA and NELF) whereas less stably paused, more processive genes have a strong +1 nucleosome signal that causes more transient pausing. Do the authors find something significantly different in the mammalian system? Can they show by heatmap the relationship between the level of pausing and the occupancy level or turnover of the +1 nucleosome and analyze this quantitatively to demonstrate better that the +1 nucleosome might be a static barrier to mammalian PolII?

---

## [Author Response]

*As you will gather from the full reviews included below, there was a clear disparity in the initial assessment of your manuscript. Whereas Reviewers 1 and 2 observed the various strengths of your work and found it appropriate for publication in eLife, Reviewer 3 pointed out that several key observations into the role of Chd1 in nucleosome remodeling at distinct genomic locations described in your manuscript had been previously established in the literature. This led Reviewers 1 and 2 to re-review your manuscript to determine to what degree the novel aspects in your work constituted a significant advance in the field. After this process, Reviewers 1 and 2 agreed that overall the paper ranks as very good and that there is new information in the manuscript*.

*Upon further discussion, we would like to encourage a resubmission of a revised manuscript that not only thoroughly addresses the technical and data interpretation concerns raised by Reviewer 3, but also is rewritten in a way that better emphasizes what was known and what is novel in this paper about the role of Chd1 in nucleosome position and exchange. We also encourage addition of any new available data that would further advance our understanding of the molecular mechanism of action of Chd1*.

Thank you for taking the time to thoughtfully review our manuscript. We have addressed the concerns in the revision and have included a detailed description in response to the comments below.

We believe that our initial submission did not emphasize the novel nature of our method for high resolution mapping of Chd1 and PolII. This technique is based upon the MNase digestion of crosslinked chromatin, limited sonication for quantitative ChIP recovery, and the sequencing of protected fragments. The development of the technique was essential for identification of Chd1 binding to the surface of nucleosomes, as previously implicated in vitro. In addition, this technique is highly adaptable, only requiring a simple modification to the regularly used ChIP-seq protocol and uses existing sequencing platforms. In the resubmission, we have provided further details of the method as a resource for the scientific community.

As detailed below, we have rewritten the manuscript to address previous work on the role of Chd1 in yeast. In brief, from the point of view of this study, yeast differ in two main ways: (1) yeast have the TSS embedded within the +1 nucleosome, whereas in metazoans the entry site for the +1 nucleosome is ∼50 bp downstream; (2) unlike the situation in yeast, the vast majority of metazoan genes exhibit high levels of PolII accumulation near the promoter. As such, we set out to investigate Chd1 function in mammals. Our motivation was spurred on by the observation that Chd1 is required for metazoan specific phenomena – maintenance of pluripotency and reprogramming, for which we provide possible mechanistic insights. Also, understanding the regulation of nucleosome turnover in mammals is vital, for example with the recent observation that the widely used anti-cancer drug doxorubicin increases turnover, consistent with the tumor suppressor role of mammalian Chd1.

We find a number of unique changes to chromatin that are not found in yeast: (1) the +1 nucleosome occupancy is decreased in the absence of Chd1 activity; (2) Chd1 primarily drives histone turnover at the promoter, compared to the relatively weak effect seen in the gene body. Work in yeast has not yielded a clear link between changes in chromatin and transcription upon loss of Chd1. Here we show that Chd1 occupancy and activity is linked to transcription, with Chd1 responsible for essentially all of PolII-dependent histone turnover at the promoter and removes the barrier to PolII promoter escape.

Reviewer 1:

*My only minor concern is the creation of the RNAPII 'stalling index' in*
Figure 6*, which is basically equivalent to the RNAPII 'pausing index' used by many reports in the literature. There is a minor technical difference in how Skene et al. calculated their index (i.e., using Ser2-phospho-RNAPII as denominator instead of total RNAPII) which I don't mind, but I don't think there is need to introduce 'stalling' here, I'll be happy with 'pausing'*.

The reason that we defined the term as ‘stalling index’ is for the subtle differences in the definition of stalled vs paused polymerases, as defined by John Lis (PMID: 19056941; supplemental p13). In brief, paused polymerases specifically are elongation competent, whereas the concept of a stalled polymerase is an inclusive term for all polymerase density (paused, backtracked and arrested). Our metric makes no assumption concerning the elongation potential of the polymerase and therefore we favor the term ‘stalling index.’

Reviewer 2:

*The authors are experts in chromatin biology, and have developed several assays that enable different attributes of promoter chromatin to be analyzed. Here, they explore the impact of Chd1 remodeler on promoter chromatin. This issue has been addressed previously in ES cells at lower resolution, where promoter association was observed, with Chd1 loss leading to the acquisition of heterochromatin*.

*The paper starts with the application of methods developed in the Henikoff lab to examine both Chd1 and RNA PolII in MEFs. This involves the separate examination of larger and smaller fragments derived from limited MNase digestion of crosslinked chromatin, following chromatin IP (ChIP). The larger fragments represent linkage of the factor to nucleosomes, whereas the smaller represent the factor's footprint. Support is provided that smaller fragments with PolII better link to the TSS*.

As mentioned above, the work presented here actually uses a novel technique that was specifically developed to address the distribution of Chd1. Typically our lab has favored native ChIP conditions. However, in mammalian cells we found that under native ChIP conditions only 14% of Chd1; 10% of PolII and ∼25% of histone H3 could be extracted, raising concerns that perhaps native ChIP would only investigate a specific genomic compartment of Chd1 binding. In comparison, sonication is typically used to both fragment chromatin and solubilize proteins in crosslinking ChIP, which results in low resolution and variable recovery. Here, we present the novel combination of formaldehyde crosslinking, MNase digestion to yield protected fragments and limited sonication to solubilize near 100% of Chd1 and chromatin. Through this technique we are able to generate high resolution data of Chd1 and PolII distributions. This was vital to observe the site of action of Chd1 on nucleosomes, as had been previously indicated in vitro (PMID: 16086025 and 16518397) and the stalled PolII in front of nucleosomes upon loss of Chd1 activity. This highly transferable technique has broad appeal to the scientific community, as it requires no special tool generation and uses current sequencing technology. The manuscript has been expanded to emphasize the novel nature of this technique, and Figure 1 now includes further explanation and details of the complete solubilization of chromatin obtained with this method (Results section referring to Figure 1; Discussion).

*For Chd1, the authors examine both an epitope-tagged WT and a catalytic mutant. With both Chd1 derivatives, localization to the +1 nuc was observed, however the catalytic mutant showed additional occupancy of proximal downstream nucleosomes. This data was nicely done and described. The authors then explored the impact on nucleosome occupancy and positioning following expression of the dominant-negative Chd1 derivative. Here, they claim a loss in nucleosome occupancy without a change in positioning. I find the effect very small overall, even when PolII levels are considered, but much more clear in the case where the highest quintile of Chd1 occupancy is examined. The authors note a difference in impact here in MEFs on the +1 nuc, versus in budding yeast where no change is observed following Chd1 loss, which I found interesting*.

In the revision, we now emphasize this difference, especially as it implies a fundamental difference between yeast and metazoans in how transcription proceeds through chromatin at the promoter.

*I also found the correlation of Chd1 with 'open' promoters interesting. The authors then apply an alternative shRNA approach to knock down Chd1, and observe similar (and even somewhat stronger) results*.

*As prior approaches provide steady-state measurements, the authors then apply the CATCH-IT technique to explore dynamics. First, they show that turnover is high at locations flanking the TSS. The authors then explore the impact of expression of the dominant negative. The authors emphasize a significant difference at the locations flanking the TSS, and in the gene body. However, this is really confined to the nucleosome directly on, or adjacent to, the TSS; here it is clear and interesting, however, regarding the gene body, I found the difference with the dominant negative apparent, but the scale/magnitude quite small*.

We agree that the gene body turnover difference is small in absolute magnitude, which emphasizes that the biggest effect by far of mammalian Chd1 is at promoter proximal nucleosomes. This is fundamentally different from what is seen in yeast, where suppression of turnover in the gene body appears to be the primary role of yChd1 (PMID: 22922743; 22807688). We then extend this observation of mammalian Chd1 driving turnover and show that is responsible for essentially all of PolII directed nucleosome turnover, indicating its fundamental role in transcription through promoter proximal chromatin. The text has been modified to emphasize this novel role of Chd1, not evident from studies in yeast.

Reviewer 3:

*This manuscript investigates the role of Chd1 in transcription. Using MNase-seq, PolII ChIP-seq and CATCH-IT in MEF cells expressing WT or a dominant negative form of Chd1, the authors conclude that Chd1 is a factor required for the RNAP to overcome the barrier of the +1 nucleosome and relieve stalling of RNAP, whereas in the gene body it suppresses nucleosome turnover. There are some potentially good ideas in this manuscript, but the data do not significantly extend what we know from previous work and the data suffers from insufficiently rigorous analysis*.

*These results are similar to work from a number of labs (Workman, Tamkun, Hartzog, Rando, Kornberg), showing that Chd1 localizes at promoters and within gene bodies where it facilitates transcription by altering nucleosome occupancy and turnover/histone replacement, especially at highly active genes. In particular, the data is reminiscent of Radman-Livaja et al. (PLOS Genetics 2012), where it is shown in yeast that loss of Chd1 activity leads to slower turnover of nucleosomes over the promoter and faster turnover in gene bodies and near 3' ends*.

We are not disputing the importance of work that has been done with regards to Chd1 function in yeast. There are, however, key differences between our results in mammals and yeast, which likely reflects differences in promoter function and chromatin architecture. In addition, our work presents the novel finding that Chd1 functions to overcome the nucleosomal barrier to transcription, which has not been previously shown.

In yeast, studies have focused on Chd1 suppressing histone turnover in the gene body (PMID: 22807688; 22922743), which we also observed in mammals, and further show that Chd1 recruitment to the gene body closely follows elongating PolII density. There are, however, fundamental differences in both transcription and chromatin context at the promoter between mammals and yeast. First, in yeast the transcription start site is embedded within the +1 nucleosome (PMID: 22258509), implying that the +1 nucleosome must be mobilized prior to pre-initiation complex formation. In contrast, in mammals the TSS is embedded within the nucleosome depleted region and the +1 nucleosome entry site found on average ∼50 bp downstream. Second, in mammals the vast majority of genes display PolII accumulation at the promoter region (91% of genes have a traveling ratio >2; PMID: 20434984), whereas in yeast the PolII density is more evenly distributed along the length of the gene (PMID: 21248844; 22258509). This suggests a fundamental difference in the mechanism of PolII transit through the promoter proximal nucleosomes, which is supported by the identification of specific role of mammalian Chd1 at the promoter in driving turnover. In contrast to yeast where the magnitude of change in turnover at the promoter is approximately the same as in the gene body (PMID: 22807688; 22922743), the absence of mammalian Chd1 activity leads to a much greater reduction in promoter histone turnover. We present the novel finding that mammalian Chd1 is responsible for almost all PolII dependent turnover at the promoter and loss of this activity results in further accumulation of PolII at the promoter, a metazoan specific phenomenon. The previous yeast studies have not identified a consequence for transcription upon the loss of turnover at the promoter in the absence of Chd1. As such, our study identifies novel roles of Chd1 in terms of transcription related chromatin changes and places Chd1 as vital in the manipulation of metazoan specific chromatin structure.

The text has been altered to include the yeast studies describing the role of Chd1 in turnover and emphasize that mammalian Chd1 is primarily functioning to drive turnover at the promoter in contrast to yeast (Results section entitled “Chd1 has opposing roles in regulating nucleosome turnover at the promoter and the gene body”).

*The authors emphasize the idea that Chd1 does not remodel promoter nucleosomes in yeast, but I don't understand why (they cite a Science paper from the Owen-Hughes lab that focused on nucleosome positioning). In fact that paper showed lower nucleosome occupancy near promoters in remodeling deficient cells, in total agreement with the current manuscript*.

This study (PMID: 21940898; Figure 1) showed that loss of Chd1 did not affect +1 nucleosome occupancy, but instead resulted in a loss of the downstream nucleosome occupancy, and an increase in occupancy at the −1 nucleosome. However, our biggest effects are observed at the +1 nucleosome: for loss of occupancy, loss of turnover, high resolution Chd1 binding and PolII stalling. This difference is likely indicative of the differing mechanism of transcription through chromatin at promoter nucleosomes in metazoans. Furthermore, we go on to describe the transcriptional barrier to metazoan PolII transit formed by the +1 nucleosome and show how Chd1 overcomes this barrier using a novel technique for the high resolution mapping of PolII. The text has been altered to provide further details of this study and to highlight the key difference that we observe here (in the Results section entitled “Chd1 activity is required to maintain nucleosome occupancy within the promoter region and the gene body”).

*Furthermore, the Kornberg lab (Ehrensberger, PNAS 2011) has done nice work in S. cerevisiae showing that Chd1 is a promoter-specific chromatin remodeler, and work in S. pombe (Walfridsson et al, EMBO 2007) showed that Chd1 localized particularly to promoter regions where it helped remove nucleosomes near the transcription start site to enable transcription*.

These studies did not to identify a genome-wide link between Chd1 dependent chromatin changes and the changes in transcription. Here we have identified a unique role of mammalian Chd1 in regulating promoter nucleosome nuclear architecture in order to facilitate PolII promoter escape. These yeast studies have been acknowledged and discussed (In the Results section entitled “Chd1 activity is required for efficient promoter escape by PolII”).

The Ehrensberger study (PMID: 21646535) investigated the chromatin structure at the *PHO5* promoter and indicated that Chd1 remodeling activity removed the +1 nucleosome in vitro (denoted as N-1 in the study; the removal of the upstream nucleosome was concluded to be independent of remodeling activity). This is in contrast to the Owen-Hughes genome-wide analysis referred to above indicating no role for Chd1 in remodeling the +1 nucleosome (PMID: 21940898). This raises the possibility that Chd1 has a unique role at the *PHO5* promoter. Additionally, the Kornberg study admits that in vivo *PHO5* activation was robust in the *chd1*Δ strain, but required synergistic loss of *ISWI* to compromise gene activation. This raises the possibility of a separable function of Chd1 remodeling the yeast +1 nucleosome and ISWI-dependent transcription. In contrast, here we find absence of Chd1 activity alone has a genome-wide effect on metazoan-specific chromatin architecture and provide evidence for its vital role in overcoming the chromatin barrier to promoter escape via a novel high resolution mapping strategy.

The Walfridsson study in *S.pombe* (PMID: 17510629) indicated that mutant strains displayed increased nucleosome density at Chd1-bound promoters, the opposite of what we observe here. This may reflect the unusually short 7 bp average linker length of *S.pombe*, which is very different from that of mammals, which is ∼50 bp. Moreover, the resolution (a single 500 bp Eurogentec microarray probe per promoter) was too low to draw distinctions between promoters and flanking nucleosomes, and only half of Chd1 bound promoters showed a change. The authors went on to look for changes in steady state RNA levels and found that 580 genes were upregulated and 238 genes were downregulated, suggesting that most genes were activated by the absence of Chd1. There was statistically insignificant overlap between the genes that showed increased nucleosome occupancy at the promoter upon loss of Chd1 and the expression changes (their Figure 5), thereby failing to demonstrate a role for Chd1 in modulating promoter nucleosomes and regulating transcription. As such we felt that there was insufficient understanding of the role of Chd1 in transcription and in particular in the case of mammals, which have a different chromatin structure. Here we set out to determine the role of mammalian Chd1, and we find a strong correlation between Chd1 occupancy, changes in chromatin organization and the regulation of PolII promoter escape.

*Thus, I don't see significant novelty in the majority of conclusions presented and don't find compelling the suggested connection between Chd1 activity at promoters and the metazoan-specific phenomenon of promoter stalling of RNAP. On this point, the hypothesis that “PolII is being blocked in Chd1 mutant cells by nucleosomes despite the overall occupancy of the +1 nucleosome being lower is not clear to me. I* can *see how PolII could be blocked by an increase in nucleosome occupancy or by more static nucleosomes, but that is not what is indicated by this data. What the authors show is reduced +1 occupancy and reduced +1 turnover, which is most simply explained by nucleosomes being removed and not efficiently replaced. There is no clear data that sheds light on how this lower occupancy or failure to re-assemble nucleosomes might inhibit PolII*.

The reviewer presumes that nucleosome occupancy is correlated with turnover. However, work in yeast has shown that the highest turnover is at the promoter despite low nucleosome occupancy (PMID: 17347438). In *Drosophila*, we observed changes in turnover with no net change in occupancy in gene bodies (PMID: 2208596). Therefore, it is important to separate steady state occupancy measurements from turnover. We find that a unique role of mammalian Chd1 is to drive PolII-dependent nucleosome turnover at the promoter, indicating the presence of more static nucleosomes at the promoter region in the absence of Chd1 activity. In the absence of Chd1 activity, we find that the barrier to PolII promoter escape is increased consistent with a more static chromatin barrier. The text has been altered to clarify this point (third paragraph of the Discussion).

*Major*
*concerns*:

*1) There are a number of cases where strong conclusions are drawn based on non-quantitative analysis of normalized group-average profiles. Group averages* can *be dramatically shifted by a small subset of genes and are not a reliable way to draw generalizable conclusions about a population. In cases where the authors wish to say that Chd1 mutation 'increases' or 'decreases' signal, statistical evaluations should be performed and described clearly in the methods section. When the authors wish to say that two signals 'correlate' with each other, correlation analysis should be performed and Pearson/Spearman R squared or similar values should be presented*.

Statistical analysis has been performed and details included in the corresponding figure legends.

*2) It is really hard to interpret the data on PolII ChIP in cells expressing the catalytically dead Chd1 mutant without knowing what the effects of blocking Chd1 activity are on gene expression. The authors should at the very least select some example genes where loss of Chd1 increases the PolII stalling index and measure the nascent RNA levels at that gene by run-on or RT-PCR of intronic regions*.

We are interested in events occurring during transcription, whereas analysis of nascent chains can be complicated by co-transcriptional processing events. Therefore analyzing events that occur at the level of PolII are more relevant to the Chd1-related processes that we are studying here rather than analyzing the RNA produced downstream, especially as Chd1 has been proposed to interact with splicing factors (PMID: 18042460). In any case there is a close association between GRO-seq and PolII-Ser2-phos ChIP (PMID: 23062713), so nascent chain analysis is unlikely to provide any further insight. Therefore we favor our approach to measure ongoing transcription by genome-wide mapping of elongating PolII. Moreover we show that this reduction in elongating PolII density is more pronounced at genes with higher levels of Chd1 occupancy and is concomitant with an accumulation of PolII at the promoter, likely indicating a direct effect.

*The interpretation of the authors is that PolII is being 'blocked' at the promoter by the presence of more stable nucleosomes, but without showing that there is actually less active elongation within the gene bodies, this conclusion is very weakly supported, especially in light of work in Drosophila that shows normal Ser2-P PolII (active elongation form) levels in Chd1 mutants (Armstrong and Tamkun labs)*.

This previous observation of no change in Ser2-P PolII was based upon immunofluorescence of *Drosophila* polytene chromosomes (PMID: 18202396; Figure 4). It is unlikely that immunofluorescence would be sufficiently sensitive to small changes in overall abundance and those authors performed no quantitative analysis, stating *“While it is formally possible that transcription is affected in a subtle way, our experiments allow us to conclude that CHD1 is not absolutely required for the association of elongating PolII on chromosomes.”* The authors go on to say that “*However, since PolIIo*^*ser2*^
*levels* can *occasionally be reduced on chromosomes derived from* chd1 *mutant larvae, CHD1 activity may indirectly impact transcriptional elongation.”* The authors suggest that this observed variability with some larvae displaying reduced PolIIo^ser2^ levels may be explained through the observation of persistent Chd1 protein expression, likely as a result of maternal contribution. The authors state that “*We note that lower levels of CHD1 protein on polytenes from* chd1 *mutant larvae* can *correlate with reduced levels of PolIIo*^*ser2*^
*(data not shown)”.* Therefore, our observation of a decrease in genome-wide elongating PolII is not at all inconsistent with the results of this study.

*3) It is very surprising that the authors don't mention any of the considerable work done on the relationship between PolII and the +1 nucleosome in Drosophila. In particular, Dave Gilmour's recent EMBO paper clearly shows that there are different classes of genes that differ in the role of the +1 nucleosome. Genes with the most paused PolII lack a significant +1 nucleosome signal and are paused by separate factors (GAGA and NELF) whereas less stably paused, more processive genes have a strong +1 nucleosome signal that causes more transient pausing. Do the authors find something significantly different in the mammalian system? Can they show by heatmap the relationship between the level of pausing and the occupancy level or turnover of the +1 nucleosome and analyze this quantitatively to demonstrate better that the +1 nucleosome might be a static barrier to*
*mammalian PolII*?

Here we are investigating nucleosome dependent effects on PolII transit. NELF controls promoter proximal pausing, which occurs ∼30 bp downstream of the TSS, before the +1 nucleosome is encountered. We find that chromatin remodeler Chd1 preferentially operates on highly processive genes, which are independent of NELF regulation, to remove the nucleosome barrier to PolII. We have now altered the manuscript to include this work (last paragraph of the Results). We have also analyzed the data and find a similar situation in mammals, with processive genes having higher promoter proximal nucleosome occupancy and further show that the +1 nucleosome is more dynamic at processive genes (Figure 6). This supports the conclusion that the +1 nucleosome forms a highly dynamic barrier to transcription at processive genes.